# Binding Properties of Photosynthetic Herbicides with the Q_B_ Site of the D1 Protein in Plant Photosystem II: A Combined Functional and Molecular Docking Study

**DOI:** 10.3390/plants10081501

**Published:** 2021-07-21

**Authors:** Beatrice Battaglino, Alessandro Grinzato, Cristina Pagliano

**Affiliations:** 1Applied Science and Technology Department—BioSolar Lab, Politecnico di Torino, Environment Park, Via Livorno 60, 10144 Torino, Italy; beatrice.battaglino@polito.it; 2Department of Biomedical Sciences, University of Padova, Via Ugo Bassi 58 B, 35121 Padova, Italy; alessandro.grinzato@unipd.it

**Keywords:** Photosystem II, D1 protein, *Pisum sativum*, herbicides, optical assays, OJIP transient, molecular docking, free energy calculations

## Abstract

Photosystem II (PSII) is a multi-subunit enzymatic complex embedded in the thylakoid membranes responsible for the primary photosynthetic reactions vital for plants. Many herbicides used for weed control inhibit PSII by interfering with the photosynthetic electron transport at the level of the D1 protein, through competition with the native plastoquinone for the Q_B_ site. Molecular details of the interaction of these herbicides in the D1 Q_B_ site remain to be elucidated in plants. Here, we investigated the inhibitory effect on plant PSII of the PSII-inhibiting herbicides diuron, metobromuron, bentazon, terbuthylazine and metribuzin. We combined analysis of OJIP chlorophyll fluorescence kinetics and PSII activity assays performed on thylakoid membranes isolated from pea plants with molecular docking using the high-resolution PSII structure recently solved from the same plant. Both approaches showed for terbuthylazine, metribuzin and diuron the highest affinity for the D1 Q_B_ site, with the latter two molecules forming hydrogen bonds with His215. Conversely, they revealed for bentazon the lowest PSII inhibitory effect accompanied by a general lack of specificity for the Q_B_ site and for metobromuron an intermediate behavior. These results represent valuable information for future design of more selective herbicides with enhanced Q_B_ binding affinities to be effective in reduced amounts.

## 1. Introduction

Photosystem II (PSII) is a multi-subunit pigment-protein complex embedded in the thylakoid membranes of cyanobacteria, algae and plants. It catalyzes the light-driven water splitting reaction, which produces molecular oxygen released into the atmosphere. This reaction initiates through the absorption of solar energy by pigments of the light-harvesting system, which is then transferred as exitonic energy to the reaction center (RC) core. Here, the absorbed energy is converted into electrochemical potential energy and the water splitting occurs [1]. In all oxygenic photosynthetic organisms, the PSII RC core is composed of two homologous proteins, D1 and D2, which have five transmembrane α-helices each and bind cofactors involved in the charge separation and water splitting processes and the electron transfer chain [2]. Among these cofactors there are a cluster of chlorophyll *a* (Chl *a*) molecules referred to as P680, pheophytins (Pheo), plastoquinones (PQ), additional β-carotenes, a non-heme iron and a Mn_4_O_5_Ca cluster. The charge separation originates from the excited state of the Chl *a* P680 (P680*) within the D1 protein. The P680* reduces a Pheo molecule forming the P680^+^Pheo^−^ pair. On the electron donor side, the P680^+^ has a very high redox potential and, via the redox-active D1 Tyr161 (Y_Z_), oxidizes the Mn_4_O_5_Ca cluster in the oxygen-evolving complex (OEC) located at the C-terminus of the D1 protein. The OEC, through four consecutive oxidation steps, oxidizes two water molecules into molecular oxygen and four protons. On the electron acceptor side, the Pheo^−^ reduces the primary quinone electron acceptor, the PQ bound to the Q_A_ site of the D2 protein that can accept one electron at a time. This PQ mediates a two-step reduction of the secondary quinone electron acceptor, the PQ bound to the Q_B_ site within the D1 protein that, once reduced to plastoquinole PQH_2_, is released from PSII. To understand the mechanisms of water splitting, charge separation and electron transfer within the PSII, it has been essential to elucidate the structure at atomic-resolution of this enzyme. The first high-resolution structure was obtained at 3.8 Å through X-ray crystallography of PSII dimeric cores from the cyanobacterium *Thermosynechococcus elongatus* [3]. In the last two decades, the cyanobacterial PSII structure has been refined to 1.9 Å by X-ray crystallography [4,5] through several optimization steps (reviewed in [1]). Recently, with this technique a high-resolution (2.7 Å) PSII structure has been obtained for the eukaryotic red alga *C. caldarium* [6]. In the last few years, the explosive development of the cryo-electron microscopy (cryo-EM) technique allowed obtaining high-resolution structures of PSII also for green algae (*C. reinhardtii* at 2.7 Å [7]) and higher plants (*S. oleracea* at 3.2 Å [8], *P. sativum* at 2.7 Å [9] and at 3.8 Å [10] and *A. thaliana* at 5.3 Å [11]).

PSII-inhibiting herbicides are molecules that interfere with the photosynthetic electron transport at the level of the D1 protein, which has long been considered the herbicide-binding protein [12]. PSII inhibitors belong to different chemical families, among which the most common are phenyl-carbamates, pyridazinones, triazines, triazinones, triazolinones, uracils, amides, ureas, benzothiadiazinones, nitriles and phenyl-pyridazines [13]. Despite the diversity in the chemical structure, most of these herbicides contain a common basic chemical element, a sp^2^ hybrid carbon adjacent to a nitrogen with a lone electron pair, and act by competing with the native PQ molecule for the Q_B_ site of the D1 protein, thus blocking the electron transfer from Q_A_ to Q_B_ [14,15]. Until now, however, the molecular details of the interaction of most of these herbicides in the Q_B_ niche remain to be elucidated. Indeed, to date only one high-resolution structure of PSII binding an herbicide (terbutryn) is available from cyanobacteria [16] and none is available from algae and plants. In addition, some crystal structures of RCs binding photosynthetic herbicides are available for some anaerobic photosynthetic purple bacteria (terbutryn/*B. viridis* [17] and *R. sphaeroides* [18], and atrazine/*B. viridis* [19]) which contain the L and M subunits that, based on sequence homology, are remarkably similar to the D1 and D2 proteins [20].

Originally, photosynthetic herbicides acting on PSII were classified by the Weed Science Society of America (WSSA) and the Herbicide Resistance Action Committee (HRAC) in two distinct ways [21]. The current HRAC/WSSA classifications have been updated (https://www.hracglobal.com; https://wssa.net, assessed on 15 March 2021) and here we will refer to this new version. Both systems now classify the photosynthetic herbicides in two groups, group 5 and group 6, which reflect the different sites of action of the herbicides on the D1 protein (Table 1). These classifications rely on studies either of naturally occurring herbicide-resistant plants [22,23,24] and mutational variants at the acceptor side of PSII [14,25] or of FTIR spectroscopy and docking analyses [26,27,28]. Particularly important are the early characterization studies of atrazine-resistant weed biotypes [29,30] and of several D1 mutants generated in cyanobacteria, algae and plants exhibiting herbicide-resistance (reviewed in [14]). These studies highlighted in particular the mutation of Ser264 to Gly, but also to Ala, Asn, Pro and Thr as responsible for the resistance of the organisms against certain classes of herbicides like triazines, ureas and triazinones, while retaining the sensitivity towards other herbicides such as the phenolic ones (now classified as nitriles). The observed different behavior, together with the peculiar chemical features of each molecule, induced Trebst to postulate for the latter photosynthetic herbicides a binding area in the Q_B_ niche alternative to Ser264 and identified as His215 [31]. All together, these results suggested the existence of two groups of PSII-inhibitors depending on their preferential binding orientation within the D1 Q_B_ binding pocket: the Ser264 binders (group 5, containing ureas, amides, triazines, triazinones, phenylcarbamates, pyridazinones, uracils) and the His215 binders (group 6, containing benzothiadiazinones, nitriles, and phenyl-pyridazines). 

In this work, we investigated the inhibitory effect on plant PSII of five commercial and widely used photosynthetic herbicides belonging to different chemical classes, namely diuron (i.e., DCMU (urea), the most commonly used inhibitor in photosynthesis research), metobromuron (urea), bentazon (benzothiadiazinone), terbuthylazine (triazine) and metribuzin (triazinone) (Table 1). We combined in vitro experiments, via OJIP chlorophyll fluorescence kinetic measurements and PSII activity assays performed on thylakoid membranes isolated from pea plants, with in silico molecular docking analyses, using the highest resolution structure available for PSII from the same plant. Both approaches showed for terbuthylazine, metribuzin and diuron the highest affinity for the Q_B_ site of the D1 protein, with the latter two molecules able to form hydrogen bonds with His215. Conversely, they revealed for bentazon the lowest PSII inhibitory effect accompanied by a general lack of specificity for the Q_B_ site and for metobromuron an intermediate behavior. Moreover, each of these PSII inhibitors displayed a specific molecular interaction pattern within the Q_B_ site of the plant D1 protein that goes beyond the chemical class of belonging. Considering the dramatically increasing use of pesticides for weed control in agriculture, the knowledge derived from this combined computational and experimentally validated study could be helpful for future design of more selective photosynthetic herbicides with enhanced Q_B_ binding affinities to be effective in reduced amounts.

## 2. Results and Discussion

### 2.1. Structural Conservation of the D1 Protein and Q_B_ Binding Site among Oxygenic Photosynthetic Organisms

In all oxygenic photosynthetic organisms, comprising cyanobacteria, algae and plants, PSII works as a water–plastoquinone oxidoreductase, which extracts electrons from water and shuttles them to a quinone pool made of Q_A_ and Q_B_ molecules. In the last decades, the several high-resolution PSII structures solved provided solid bases for elucidating the electron transport occurring within the PSII complex. It is well known that PSII-inhibiting herbicides cause the interruption of this electron transport between Q_A_ and Q_B_ [14,15]. So far, however, there is a complete lack of information at molecular level on the interaction of the PSII inhibitors binding the Q_B_ site within the D1 protein in plants, which is the real target of the PSII-inhibiting herbicides commercially used for weed control. Indeed, the molecular detail of this interaction in oxygenic photosynthetic organisms is limited to an atrazine-type of PSII-inhibiting herbicide, which was co-crystallized with the PSII dimeric core isolated from the cyanobacterium *Thermosynecococcus elongatus* [16]. In this work, the cyanobacterial PSII structure binding the terbutryn herbicide has been obtained at 3.2 Å, allowing elucidating the interaction network of this herbicide within the Q_B_ niche of the D1 protein. 

A general high conservation of the structure of the RCs among photosynthetic organisms is recognized [32]. Within the RC, the D1 protein consists of five transmembrane α-helices, named A–E (Figure 1A), which are in close proximity to the D2 protein. To investigate the structural conservation of the Q_B_ binding site of the D1 protein among oxygenic photosynthetic organisms, the amino acid sequences of the D1 protein retrieved from the plant PSII structures were compared to their counterparts from algae and the cyanobacterium *T. elongatus* with the terbutryn bound (Figure 1A). The alignment showed a high conservation among the D1 protein sequences of algae and plants with respect to the cyanobacterial counterpart, ranging respectively from 92.3% to 94.8% similarity (i.e., 92.3% for *C. reinhardtii*, 93.3% for *C. caldarium*, 94.5% for *A. thaliana*, 94.8% for *S. oleracea* and *P. sativum*). The plant PSII structures, with respect to the cyanobacterial counterpart, revealed also a high conservation of the pigments (four Chl *a*, two Pheo and one β-carotene) and other ligands (e.g., a non-heme iron coordinated by the D1 His215 and His272) bound to the D1 amino acid backbone, as shown in Figure 1B for the D1 protein of *P. sativum*. Among the ligands, it should be noted that in all the PSII structures considered the plastoquinone Q_B_ was entirely solved only in the cyanobacterial one, whereas it was absent from that of eukaryotes with the exception of *P. sativum* [9,10], where only the head and a truncated tail is visible (Figure 1C). Unlike the other cofactors that are tightly bound to the PSII, the Q_B_ molecule displays a substrate-like behavior, and this might be the reason of the general difficulty to assign its density in the PSII structure. Comparing all these structures, it is evident that the Q_B_ niche located in the D1 region formed by the D and E α-helices and the DE connecting loop, which in eukaryotes consists of at least 65 amino acids spanning from Phe211 to Leu275, contains several hydrophobic amino acids, among which Phe211, Leu218, Phe255, Phe265, Leu271, Phe274 and Leu275 (Figure 1A). These amino acids accommodate the apolar head and tail of the PQ (Figure 1C). At the bottom of the Q_B_ pocket, the polar groups of His215 and/or of Ser264 or Phe265, depending on the organism/structure considered, form hydrogen bonds with the keto-oxygens of the PQ head (Figure 1C). All these amino acids are highly conserved among the representatives of cyanobacteria, algae and plants considered (Figure 1A), thus suggesting that the Q_B_ site is conserved as well among the oxygenic photosynthetic organisms. Based on structural data from the cyanobacterium *T. elongatus*, some of the residues of the Q_B_ niche involved in binding the native PQ molecule [33] are the same binding the herbicide terbutryn [16] (Figure 1A). Despite the lack of herbicide-bound PSII structure in plants, these observations suggest that these highly conserved amino acid residues in the Q_B_ site may play a key role in binding PSII-inhibitors also in plants. 

### 2.2. Estimation of the Herbicide Binding Affinity for the Q_B_ Binding Site of the D1 Protein by Photochemical and Fluorescence Assays of Photosystem II Inhibition 

To compare the inhibitory effect of the herbicides diuron, bentazon, metribuzin, metobromuron and terbuthylazine on the electron transfer efficiency of the pea thylakoid membranes, we performed the photochemical assay of 2,6-dichlorophenolindophenol (DPIP) photoreduction which relies on the Hill reaction [34]. In this test, the PSII activity can be measured as electron transfer rate in saturating light conditions from water to the “Hill oxidant” DPIP (i.e., the Hill reaction), which acts as acceptor of electrons from the Q_B_ site of the PSII D1 protein during the light-induced oxidation of water. 

The inhibition curves of DPIP photoreduction in thylakoid membranes for the five herbicides and the corresponding concentrations inhibiting the 50% (I_50_) of the DPIP photoreduction are shown in Figure 2A and Table 2, respectively. The inhibition curves of thylakoids treated with diuron, terbuthylazine and metribuzin were similar and showed a higher decrease of electron transfer rate at lower concentrations of herbicide with respect to that of bentazon and metobromuron (Figure 2A). The identification of the I_50_ value, corresponding to the molar concentration of herbicide producing half saturation of the binding sites (equal to the PSII-herbicide dissociation constant), allows estimating the affinity of the PSII complex for the different herbicide inhibitors. In this context, where a lower I_50_ value indicates a higher affinity for the herbicide, pea PSII showed an affinity for diuron, terbuthylazine and metribuzin more than one order of magnitude higher compared to bentazon and metobromuron (Table 2).

The chlorophyll *a* fluorescence OJIP transient is correlated with the primary photochemical reaction of PSII and its study allows assessing the effect of several abiotic stress factors, including herbicide exposure, on the PSII electron transport chain [36,37]. Dark adapted samples illuminated by a continuous saturating light show a characteristic polyphasic fluorescence induction curve with a first fast rise, from the origin (O), corresponding to the minimum fluorescence (F_0_) to an intermediate step (J) at 2 ms, followed by a second slower rise through an intermediate step (I), at 30 ms, to the maximum peak (P), corresponding to the maximum fluorescence (F_M_) attained within 1 s (see Figure A1). These two rises are known respectively as the photochemical phase (O-J) and the thermal phases (J-I-P) of the fluorescence curve. F_0_ reflects the condition in which the Q_A_, as well as the PQ pool, are in the oxidized state, while F_M_ the condition in which all Q_A_ molecules are in the reduced state Q_A_^−^. The relative variable fluorescence at the J step, V_j_ = (F_2ms_ − F_0_)/(F_M_ − F_0_), represents the relative amount of reduced Q_A_ ([Q_A_^−^]/[Q_A_^−^ + Q_A_]) at 2 ms. This parameter, together with the derived 1–V_j_, is used to detect and estimate the strength of PSII inhibitors displacing the secondary acceptor quinone Q_B_ from its binding pocket in the D1 protein [38]. Indeed, when an herbicide inhibitor interferes with the PSII electron transport chain, changes in the slope and shape of the fluorescence emission curve, such as the alteration of the J peak [39] are recorded (see Figure A1). Moreover, the derived parameter 1–V_j_, which estimates the rate of re-oxidation of the Q_A_^−^ with respect to its reduction, varies proportionally to the herbicide concentration [40] and thus can be used to assess whether the herbicide exerts inhibition between the Q_A_ and the Q_B_ site. 

To estimate the strength of diuron, bentazon, metribuzin, metobromuron and terbuthylazine in displacing the native Q_B_ from its binding pocket in the D1 protein, we performed the OJIP measurements on isolated thylakoids treated with increasing concentrations of the five herbicides in comparison with untreated thylakoids (Figure A1) and then we calculated the 1–V_j_ parameter. The PSII inhibition was expressed as percentage of residual 1–V_j_ of herbicide-treated thylakoids with respect to the untreated ones. The dose-response curves for the different herbicide molecules and the concentrations inhibiting the 50% (I_50_) retrieved from the 1–V_j_ parameter for each herbicide are shown in Figure 2B and Table 2, respectively. The inhibition curves of thylakoids treated with diuron, terbuthylazine and metribuzin were similar and showed a higher decrease of variable fluorescence at lower concentrations of herbicide with respect to that of bentazon and metobromuron (Figure 2B). By comparing the I_50_ values derived from this fluorescence test, it is evident that diuron, terbuthylazine and metribuzin have an affinity for the Q_B_ site of more than one order of magnitude higher than bentazon and metobromuron (Table 2).

The photochemical assay with DPIP and the OJIP fluorescence test have been widely used to measure the inhibition of the electron flow within PSII by photosynthetic herbicides and their binding affinities [38,39,41,42]. The results obtained in our comparative study with the two experimental approaches revealed two distinct behaviors of the five selected photosynthetic herbicides on pea thylakoid membranes (Figure 2). Terbuthylazine and metribuzin affected the DPIP photoreduction rate and the OJIP fluorescence transient in a way very similar to that of diuron. The I_50_ retrieved from our experiments (Table 2), which is an estimate of the herbicide binding affinity for the Q_B_ site in the D1 protein of the PSII, was 7–8 × 10^−8^ M for diuron (urea class), in accordance with values measured in previous works on thylakoids from other species [25,29,43,44]. The I_50_ for terbuthylazine (triazine class) and metribuzin (triazinone class) was slightly higher, in the order of 1–2 × 10^−7^ M, in accordance with values in the range 10^−7^–10^−8^ M measured on different plant material for the same molecules [25,29] or other molecules of the triazine class, such as atrazine [43] and terbutryn [29]. Conversely, the inhibition trends of bentazon and metobromuron moved away from the rest of the tested molecules. Indeed, the I_50_ of bentazon (benzothiadiazinone class) and metobromuron (urea class) was about two order of magnitude higher compared to that of diuron, terbuthylazine and metribuzin. This result is in accordance with previous measurements on different photosynthetic material treated with bentazon [29,45]. Despite a common inhibition of the electron transfer at the level of the Q_B_ site of the D1 protein by all the tested herbicides, that might be related to the presence of a common structural “essential element”, a sp^2^ hybrid carbon adjacent to a nitrogen with a lone electron pair [15], the different binding affinity observed suggests a distinct interaction interface and/or orientation within the Q_B_ pocket for the members of the two groups of herbicides, one made by diuron, terbuthylazine and metribuzin and the other by bentazon and metobromuron. This could be due to the chemical diversity of each herbicide, in terms of specific side chain and steric hindrance, which goes beyond the chemical class of belonging.

### 2.3. Molecular Docking Study of the Interaction of Herbicides in the Q_B_ Binding Site of the D1 Protein

Molecular docking is a widely used computational tool for in silico prediction of the binding affinities and modes of protein-ligand interaction [46]. To describe the structural basis of interaction between the selected herbicides and the Q_B_ binding pocket of the D1 protein in plants, we performed molecular docking analyses with the docking software Audodock Vina (La Jolla, CA, USA) [47] using the PSII structure with the highest resolution available in plants (PDB: 5XNL from *P. sativum*). All the most stable conformations selected according to the docking software internal scoring function were in the Q_B_ binding pocket, forming non-polar interactions with Met214, Leu218, Phe255 and Leu271. Bentazon and metobromuron, the herbicides displaying in vitro lower affinity (Table 2), in the docking analyses took different orientations, due to the large number of possible binding sites (Figure 3), whereas diuron, metribuzin and terbuthylazine, the herbicides with in vitro higher affinity (Table 2), in the docking analyses preferred to orient towards His215 (Figure 3). In particular, diuron formed one hydrogen bond between its carbonyl group and the ND1 of His215 (Figure 3A) and metribuzin formed two hydrogen bonds between its amino group or its carbonyl group and the ND1 of His215 (Figure 3B).

The calculated solvation free energy (ΔG_int_) and free energy of assembly dissociation (ΔG_diss_) (Table 3) showed that diuron has the strongest binding ability, followed by terbuthylazine and metribuzin, which displayed similar values, while metobromuron showed a lower binding affinity compared to the previous three molecules. For bentazon, both the interaction and dissociation energies were close to zero (Table 3), indicating a lack of specificity for the Q_B_ site. Similarly, the maximum deviation from the optimal conformation (i.e., root-mean-square deviation, RMSD) of diuron, terbuthylazine and metribuzin were similar to each other (respectively 5.64 Å, 7.02 Å and 5.85 Å) and smaller than the ones of metobromuron (21.45 Å) and bentazon (30.29 Å), showing once again a higher affinity of the first three compounds over the second two ones. To validate the docking procedure adopted, the same docking simulation was performed also with the molecules diquat and paraquat, two herbicides belonging to the pyridinium class that act as Photosystem I inhibitors [13] and are not able to bind into the Q_B_ pocket, and the head of the native plastoquinone present in the *P. sativum* deposited structure (PDB: 5XNL). As expected, diquat and paraquat did not fall into the Q_B_ binding pocket and distributed themselves randomly into the docking region (RMSD respectively of 39.52 Å and 35.23 Å), conversely the plastoquinone suited perfectly into the Q_B_ pocket (RMSD of 4.23 Å) forming the same hydrogen bond present in the deposited structure (PDB: 5XNL from *P. sativum*) between a keto-oxygen of its head and the ND1 of His215.

All these in silico findings suggest that in pea plants His215 is the principal binding target for the herbicides that showed higher affinity for the Q_B_ pocket, i.e., diuron, metribuzin and terbuthylazine, irrespective of the original chemical class of belonging. Moreover, the formation of single or double strong hydrogen bonds with His215 seems to play a key role in stabilizing these molecules within the plant D1 Q_B_ pocket. These results highlight the importance of the D1 His215 not only for binding of the native PQ molecule (e.g., see PDBs 5XNL for plants, 6KAC for green algae and 4V62 for cyanobacteria) but also of exogenous PSII-inhibiting herbicides, as revealed by structural studies [16] and molecular modeling and dynamics studies [40]. Notably, His215 is involved in the molecular bridge Q_A_−D2 His214−non-heme iron−D1 His215−Q_B_. For this reason, the orientation towards or formation of H-bond(s) with His215 of the herbicides displaying the higher Q_B_ binding affinity in the docking analyses might suggest herbicide-induced conformational changes in this region, possibly pivotal to exert their superior inhibition of the photosynthetic electron flow between Q_A_ and Q_B_. To verify this hypothesis, it would be helpful to perform further in silico analyses on the D1-D2 heterodimer through molecular dynamics simulations, method already adopted to investigate the conformational changes occurring to the D1 protein during the exchange of the native PQ molecule [48].

In addition, the different behavior observed for the two tested herbicides of the urea class, i.e., diuron and metobromuron, highlighted a significant intra-class variation in the binding capacity to the Q_B_ site. This is related to the different substituents of the phenyl ring and/or tail-like of the two molecules and is in agreement with the different affinity-binding towards the Q_B_ site displayed by substituted phenylurea compounds synthesized with tail-like substituents differing in hydrophobicity, charge and length [49]. These findings and the different Q_B_ binding abilities displayed by synthesized analogues of other commercial photosynthetic herbicides (e.g., bromoxynil [28]) highlight the importance, in the future, to carry out in silico tests of a wider selection of (known and potentially acting as) PSII-inhibiting molecules by using the plant high-resolution PSII structures now available. This wider in silico analysis would contribute to better understand in plants not only the binding in the Q_B_ binding pocket of chemically different photosynthetic herbicides, but also the molecular mechanism behind the shift of the Q_A_ redox potential occurring upon herbicide binding, that in general is lowered by nitrile herbicides and increased by herbicides of the urea class [50].

## 3. Conclusions

In the last decades, several studies of randomly induced and site-directed mutagenesis performed on the D1 protein, mostly in cyanobacteria and green algae, resulted determinant to identify the amino acid residues involved in sensitivity and specificity to PSII-inhibiting herbicides [14]. Moreover, computational studies to modify the Q_B_ binding site of the D1protein through site-directed mutagenesis with the aim to improve the herbicide sensitivity have been performed by homology modelling the 3D structure of the D1 of *Chlamydomonas reinhardtii* on the X-ray structure of the cyanobacterial homolog [40,51], for which a high-resolution structure of PSII was the only available until recently. The high-resolution structures of plant PSII currently available [8,9,10,11] are of paramount importance to get insights into atomic interactions of the Q_B_ binding site of the plant D1 protein with either the native plastoquinone or exogenous herbicides, which can be investigated through docking studies, as well as molecular dynamics simulations.

This study provides the first molecular docking analysis performed in plants aimed to identify specific amino acids in the D1 Q_B_ site interacting with different PSII-inhibiting herbicides. This approach, combined with functional assays, provides new insights into the comprehension of the different affinity for the Q_B_ site and interaction patterns displayed by photosynthetic herbicides belonging to different chemical classes currently on the market worldwide. In the future, with the increase in computer power and use of more sophisticated computational tools, this kind of information could be of interest for manifold biotechnological applications. One is the design, and subsequent synthesis, of new herbicides with superior activity and greener fingerprint to reduce the dramatically increasing environment pollution [52] as well as the negative effects of weed resistance often linked to prolonged and massive use of traditional herbicides [53]. Another is the development of improved eukaryotic PSII-based biosensors suitable for the selective detection of specific classes of photosynthetic herbicides with sufficient sensitivity to meet the recommendation for the water quality guidelines, to use for in situ monitoring in aquatic ecosystems of pollutants that target specifically the photosynthetic process.

## 4. Materials and Methods

### 4.1. Plant Growth Conditions

Pea seeds, washed with 1% (*v*/*v*) sodium hypochlorite for 15 min and rinsed with distilled water, were germinated on moistened filter paper at 23 °C in the dark for 5 days. Germinated seedlings were transferred to pots and grown hydroponically in Long Ashton nutrient solution [54], which was replaced every three days. Plants were grown for four weeks in a climatic chamber (SANYO MLR-351H) at 20 °C, 60% humidity, under 8 h daylight at about 60 µmol photons m^−2^ s^−1^.

### 4.2. Thylakoid Isolation

Thylakoid membranes from 4-week-old pea leaves were isolated according to [55]. Briefly, pea leaves were grinded with a blender (Moulinex, Écully, France) in 50 mM HEPES-NaOH (pH 7.5), 300 mM sucrose and 5 mM MgCl_2_. The suspension was filtered and the filtrate was centrifuged at 1500× *g* for 10 min. The pellet was homogenized in 5 mM MgCl_2_, diluted 1:1 with 50 mM MES-NaOH (pH 6.0), 400 mM sucrose, 15 mM NaCl and 5 mM MgCl_2_ and then centrifuged at 3000× *g* for 10 min. The resulting pellet of thylakoid membranes was washed once by centrifugation for 10 min at 3000× *g* in 25 mM MES-NaOH (pH 6.0), 10 mM NaCl and 5 mM MgCl_2_. Thylakoid membranes were re-suspended in 25 mM MES-NaOH (pH 6.0), 10 mM NaCl, 5 mM MgCl_2_ and 2 M glycine betaine (MNMβ buffer) and, after freezing in liquid nitrogen, stored at −80 °C.

The chlorophyll (Chl) concentration of thylakoid membranes was determined spectrophotometrically (Lambda25 spectrophotometer, Perkin Elmer Waltham, MA, USA) after extraction in 80% (*v*/*v*) acetone according to [56].

### 4.3. Thylakoid Treatment with Herbicides

Thylakoid membranes were thawed on ice, re-suspended in the appropriate assay buffer and incubated for 10 min at room temperature in the dark. For treated samples, the assay buffer contained the herbicide at the desired final concentrations. Herbicide stock solutions were prepared at 100 mM (diuron, bentazon, metribuzin and metobromuron) or 50 mM (terbuthylazine) in pure ethanol. For control samples, the assay buffer contained no herbicide but the same volume of ethanol used for the corresponding treated counterparts. The ethanol concentrations used in all the experiments had no influence on the thylakoid electron transport activity (see Appendix A).

### 4.4. Photochemical Assay

Thylakoids samples were diluted in 50 mM HEPES-NaOH (pH 7.2), 100 mM sucrose, 5 mM MgCl_2_ and 10 mM NaCl (buffer A). The electron transfer rate of the isolated thylakoid membranes was assessed by the Hill reaction optical assay and measured from water to 2,6-dichlorophenolindophenol (DPIP) (a molecule that changes from blue to colorless while reducing) according to [57]. Briefly, the electron transfer was assessed measuring spectrophotometrically at 595 nm the DPIP photoreduction, by illuminating thylakoid membranes at a final Chl concentration of 15 µg mL^−1^ in buffer A supplemented with 5 mM NH_4_Cl in the presence of 100 µM DPIP with white light at 500 µmol photons m^−2^ s^−1^ (Fiber-Lite DC-950 lamp equipped with Liquid Light Guide Newport 77568). The isolated thylakoids were treated with increasing concentrations of herbicides, and the photoreduction rate of DPIP was calculated for herbicide-treated thylakoids and compared to that of untreated thylakoids (i.e., control). Finally, the PSII inhibition was expressed as percentage of residual electron transfer rate of herbicide-treated thylakoids with respect to the control sample.

### 4.5. Fluorescence Assay

Thylakoids samples were diluted at final Chl concentration of 2.75 μg mL^−1^ in MNMβ buffer (pH 6.0) used as assay buffer. Fluorescence induction OJIP transients were measured at room temperature, with a double modulation fluorometer FL3500 (Photon Systems Instruments, Drasov, Czech Republic). The dark-adapted samples were illuminated 1 s with continuous actinic light (2400 μmol photons m^−2^ s^−1^, emission peak at 630 nm). The first reliable point of the transient is measured at t_0_ = 0.02 ms after the onset of illumination, and was taken as F_0_. Fast fluorescence transients, thus obtained, were analyzed to estimate the rate of re-oxidation of Q_A_ with respect to its reduction, calculated as 1–V_j_ = 1− ((F_2ms_ − F_0_)/(F_M_ − F_0_)), where F_M_ is the maximum value of fluorescence recorded and F_2ms_ is the fluorescence recorded at 2 ms of illumination. The results were expressed as residual activity percentage of the herbicide-treated samples with respect to the control sample (i.e., untreated).

### 4.6. Sequence Alignment

The high-resolution PSII structures of different oxygenic photosynthetic organisms were retrieved from the Protein Data Bank (PDB) (see details in Figure 1). For each PDB entry, chain A, corresponding to the D1 protein, has been used for amino acid sequence alignment and compared to its counterpart retrieved from the UniProtKB database. The pairwise sequences have a coverage between 96% and 100%, mostly differing for some missing residues at the C-terminus. Multi sequence alignment among the UniProtKB sequences was performed in MUSCLE [58] using the default settings. The percentage of similarity between the amino acid sequence of *T. elongatus* (PDB: 4V82) and those from the other selected organisms was obtained through pairwise alignment with Emboss Needle [59] using the default setting.

### 4.7. Molecular Docking Analyses

Molecular docking simulations were carried out using Audodock Vina [47] and were guided to the Q_B_ region of the D1 protein of PSII obtained from PDB 5XNL [9], defined within a radius of 40 Å. The structure of the D1 protein was prepared by removing the plastoquinone present in the Q_B_ site, adding hydrogen atoms and charges using UCSF Chimera [60]. The herbicide 3D structures were generated, ionized, and energy minimized using Avogadro [61]; while, the plastoquinone head extracted from PDB 5XNL was used for the plastoquinone docking. Ten molecular docking simulations with different starting random seeds, that generated twenty models each, were performed for each compound. The results were scored according to Audodock Vina internal scoring function, and the root-mean-square deviation (RMSD) from the most stable conformation was calculated over the two hundred generated models. The most stable protein-compound docking models were also analyzed with PISA [62] to estimate their solvation free energy (ΔG_int_) and free energy of assembly dissociation (ΔG_diss_).

## Figures and Tables

**Figure 1 plants-10-01501-f001:**
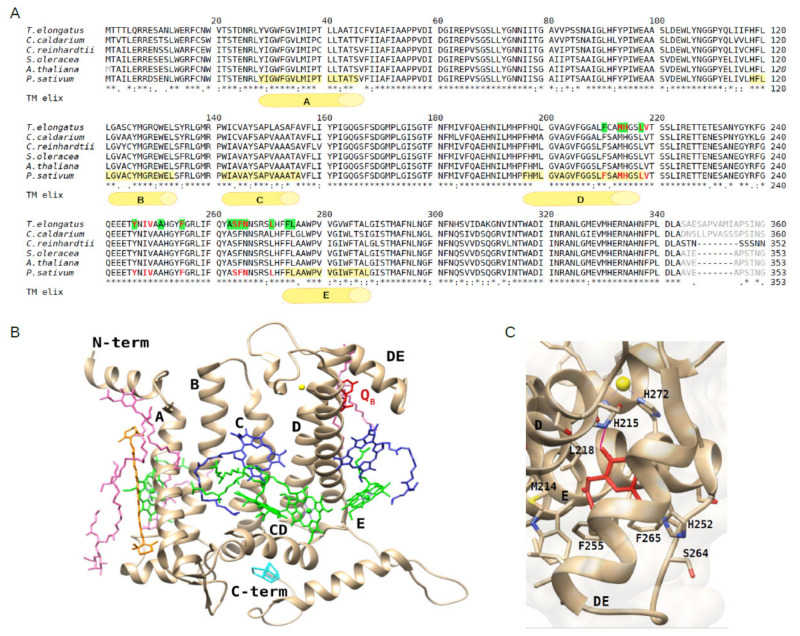
Alignment of the D1 protein sequences retrieved from high-resolution structures (chain A of each PDB entry) of representatives of cyanobacteria, algae and plants and view of the D1 protein structure and the Q_B_ binding site of *P. sativum*. (**A**) Alignment of D1 sequences from the red alga *Cyanidium caldarium* (UniProtKB O19895, PDB: 4YUU at 2.7 Å), green alga *Chlamydomonas reinhardtii* (UniProtKB P07753, PDB: 6KAC at 2.7 Å) and higher plants *Spinacia oleracea* (UniProtKB P69560, PDB: 3JCU at 3.2 Å), *Arabidopsis thaliana* (UniProtKB P83755, PDB: 5MDX at 5.3 Å) and *Pisum sativum* (UniProtKB P06585, PDB: 5XNL, at 2.7 Å) along with that from the cyanobacterium *Termosynechococcus elongatus* (UniProtKB P0A444, PDB: 4V82, at 3.2 Å). In the latter, the amino acids of the Q_B_ niche involved in the terbutryn binding are highlighted in green. Amino acids surrounding the plastoquinone Q_B_ head are colored in red in the sequences from *T. elongatus* and *P. sativum*. Amino acids not resolved in the structures are indicated in grey. The five transmembrane α-helices of the D1 protein of *P. sativum*, indicated by letters A–E, are highlighted in yellow in the corresponding sequence. (**B**) View of the D1 protein and cofactors from the structure of *P. sativum* (PDB: 5XNL, chain A). Plastoquinone Q_B_ in red, non-heme Fe in yellow, Chls in green, Pheo molecules in blue, β-carotene in orange, lipids in pink, oxygen evolving complex (OEC) in cyan. (**C**) View of the Q_B_ binding site within the D1 protein rotated with respect to the view in panel B to better show the plastoquinone head (in red), the surrounding amino acid residues and the hydrogen bond (indicated as purple segment).

**Figure 2 plants-10-01501-f002:**
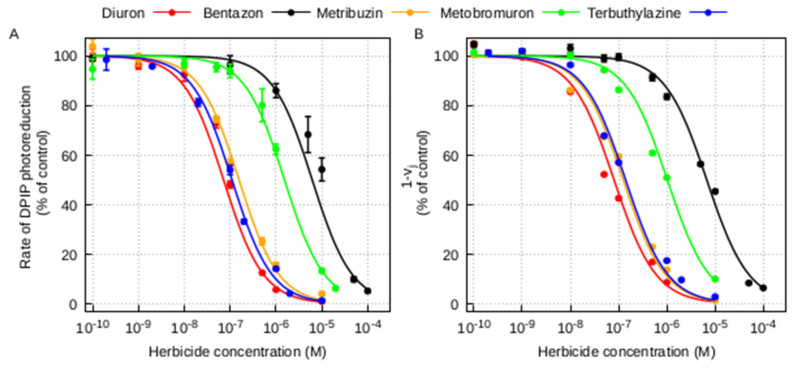
Photochemical and fluorescence assays of Photosystem II inhibition in pea thylakoid membranes by different photosynthetic herbicides. (**A**) Inhibition of DPIP photoreduction. Points are mean of at least three measurements, bars are standard errors. The experimental points were fitted using the Langmuir adsorption isotherm (R^2^ = 0.982, R^2^ = 0.960, R^2^ = 0.996, R^2^ = 0.994, R^2^ = 0.997 for the fitted inhibition curves with diuron, bentazon, metribuzin, metobromuron and terbuthylazine, respectively). (**B**) Inhibition of the variable fluorescence 1-V_j_ calculated from corresponding OJIP fluorescence transients. Points are mean of at least four measurements, bars are standard errors. The experimental points were fitted using the Langmuir adsorption isotherm (R^2^ = 0.989, R^2^ = 0.994, R^2^ = 0.993, R^2^ = 0.991, R^2^ = 0.993 for the fitted OJIP curves with diuron, bentazon, metribuzin, metobromuron and terbuthylazine, respectively). All the fittings were performed according to [35].

**Figure 3 plants-10-01501-f003:**
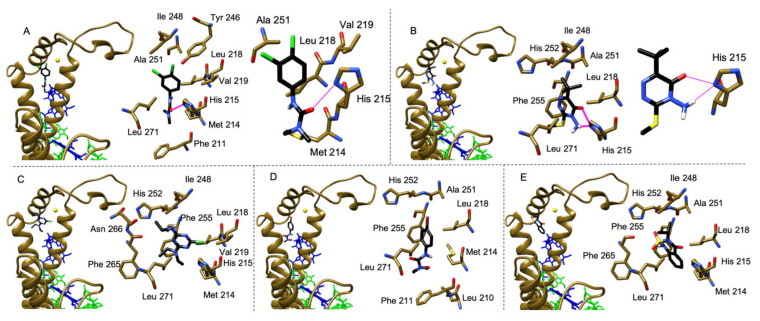
Molecular docking analysis of the interaction between the selected herbicides and the Q_B_ binding site of the D1 protein (chain A, PDB: 5XNL). Binding geometry of diuron (**A**), metribuzin (**B**), terbuthylazine (**C**), metobromuron (**D**) and bentazon (**E**). In panels (**A**–**E**), on the left view of the herbicides in the Q_B_ binding pocket followed by an enlarged view of the same region showing the amino acids in close contact (<4 Å) to each herbicide. In panels (**A**,**B**), on the right enlarged views of the same Q_B_ binding pocket region with a different orientation to show details of the hydrogen bonds formed by the herbicides diuron (**A**) and metribuzin (**B**) with His215. The herbicide molecules are colored in black, the D1 protein and cofactors are colored as in Figure 1B,C, hydrogen bonds are indicated as purple segments.

**Table 1 plants-10-01501-t001:** Herbicides used in this study.

Herbicide	Chemical Class	Classification (HRAC/WSSA)/Site of Action	Molecular Structure
Diuron	Urea	5/Ser264	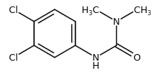
Metobromuron	Urea	5/Ser264	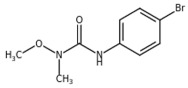
Terbuthylazine	Triazine	5/Ser264	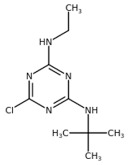
Metribuzin	Triazinone	5/Ser264	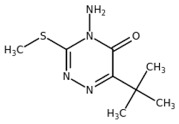
Bentazon	Benzothiadiazinone	6/His215	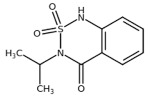

**Table 2 plants-10-01501-t002:** I_50_ values for different photosynthetic herbicides determined from PSII inhibition experiments through DPIP photoreduction (Figure 2A) and OJIP fluorescence transient (Figure 2B) assays. The I_50_ values and the corresponding standard errors were computed by fitting the experimental data using the Langmuir adsorption isotherm according to [35].

Herbicide	DPIP Photoreduction	OJIP Fluorescence
	I_50_ (M)	I_50_ (M)
Diuron	7.18 × 10^−8^ ± 1.53 × 10^−9^	8.02 × 10^−8^ ± 5.67 × 10^−10^
Terbuthylazine	1.10 × 10^−7^ ± 2.75 × 10^−9^	1.39 × 10^−7^ ± 1.60 × 10^−9^
Bentazon	6.16 × 10^−6^ ± 3.89 × 10^−7^	6.70 × 10^−6^ ± 8.30 × 10^−8^
Metribuzin	1.56 × 10^−7^ ± 5.95 × 10^−9^	1.24 × 10^−7^ ± 1.13 × 10^−9^
Metobromuron	1.53 × 10^−6^ ± 7.29 × 10^−8^	1.02 × 10^−6^ ± 1.03 × 10^−8^

**Table 3 plants-10-01501-t003:** Values of solvation free energy (ΔG_int_) and free energy of assembly dissociation (ΔG_diss_) of the most stable protein-herbicide models obtained by molecular docking simulations.

Herbicide	Molecular Docking	
	∆G_int_ (kcal/mol)	∆G_diss_ (kcal/mol)
Diuron	−2.2	1.4
Terbuthylazine	−1.4	1.5
Bentazon	−0.5	−0.1
Metribuzin	−1.5	1.1
Metobromuron	−1.1	0.7

## Data Availability

Datasets analyzed supporting reported results can be provided on request by the corresponding author.

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
