# Peer review of "Binding Properties of Photosynthetic Herbicides with the QB Site of the D1 Protein in Plant Photosystem II: A Combined Functional and Molecular Docking Study"

_plants, 2021, doi:10.3390/plants10081501_

Round 1
Reviewer 1 Report
Overview:
This manuscript uses some recently solved high resolution PSII structures from pea plants to investigate the molecular mechanisms for several classical PSII inhibitor herbicides. Overall, the research was well designed, completed and reported in this manuscript. I thank the authors for the obvious time and energy spent in preparing a well written manuscript. This work adds some interesting details to what is already known about the effects these herbicides have on the photosystem, and potentially provides clues into how additional inhibitors could be designed in the future. Below are two more detailed comments, one just to make a figure easier to read.
Detailed comments:
Line 191: This is picky, but the blue highlight is really hard to find on the computer screen, perhaps could the highlight be made a little higher contrast?
Line 258: I found the Fig A1 very helpful in understanding the results, could the authors include supplemental figures for the other herbicides tested as well?
Reviewer 2 Report
This manuscript is well written and provides in depth analysis of PS2 binding using experimental data and appropriate molecular dynamic simulations. I have some notes
-It would of been interesting to use a nitrile herbicide such as bromoxynil in the study. There have been previous studies comparing the efficacy of nitrile herbicides as influenced by subtle changes to each analog such as the paper cited here: (Cutulle MA, Armel GR, Brosnan JT, Best MD, Kopsell DA, Bruce BD, Bostic HE, Layton DS. Synthesis and evaluation of heterocyclic analogues of bromoxynil. J Agric Food Chem. 2014 Jan 15;62(2):329-36. doi: 10.1021/jf404209d. Epub 2013 Dec 27. PMID: 24354444)
-More herbicides within a PS2 herbicide group would of improved the power of this paper.
-Also more discussion is needed on how the herbicides affect the conformation of the QA−His214−iron−His215−QB system especially bentazon.
-There was minimal discussion on how the redox potential of QA is differentially effected by His215 binding herbicides and Ser264.
Reviewer 3 Report
In this study, physical and chemical analysis techniques, as well as structural simulation, were used to compare the affinity of herbicides to PSII reaction center core protein D1. It is beneficial to understand the weeding mechanism of existing herbicides and design more effective new herbicides. Because structural simulation is not my research area, I can only judge other aspects of the manuscript.
I think the main problem with this manuscripts is the writing was not concise enough. Although the author's English writing is excellent, many parts of the manuscript are repetitive, or can be greatly simplified. For example:
Line 83-106, this paragraph should be Simplified.
Line 129-147, this paragraph was similar with "Introduction", therefore should be removed.
Line 148-182, This paragraph is mainly a summary and comparison of previous studies, so it is not suitable to show in "results", but should be listed in an independent "Discussion" part.
Line 201-214, This paragraph is repeated with the “Materials and Methods”, and the description here is even more detailed than it in the "Materials and Methods". This part needs to be merged with the "Materials and Methods".
Line 241-261 the introduction of OJIP theory is beyond the necessity
Line 276- 302 Most of the content of this paragraph D is the repetition of 215-226 and 262-275.
Round 2
Reviewer 3 Report
The author has answered most of the questions and the revised version has reached the level of publication